# The Impact of Ethnic and Communication Barriers on Fatal Metabolic Emergent Management of Traumatic Injury: A Case Report

**DOI:** 10.3390/reports8040201

**Published:** 2025-10-10

**Authors:** Yutaka Furuta, Rory J. Tinker, Angela R. Grochowsky, John A. Phillips

**Affiliations:** Division of Medical Genetics and Genomic Medicine, Vanderbilt University Medical Center, DD2205 Medical Center North, Nashville, TN 37232, USA; rory.tinker@vumc.org (R.J.T.); angela.grochowsky@vumc.org (A.R.G.); john.a.phillips@vumc.org (J.A.P.III)

**Keywords:** inherited metabolic disorder, maple syrup urine disorder, metabolic emergency, plain community, traumatic injury

## Abstract

**Background and Clinical Significance**: Inherited metabolic disorders can result in fatal metabolic decompensation if not promptly recognized and treated. These conditions are common in Plain communities due to founder effects and the high prevalence of consanguinity. **Case Presentation**: We present the case of an adult Amish male with maple syrup urine disorder who sustained traumatic injuries and delayed metabolic intervention that contributed to a fatal outcome. **Conclusions**: This case highlights the critical need for increased awareness among emergency and adult care providers, especially in Plain communities, and emphasizes the importance of early multidisciplinary coordination and preparedness with metabolic resources to ensure timely, life-saving management in adult metabolic emergencies.

## 1. Introduction and Clinical Significance

Maple syrup urine disorder (MSUD) is a rare autosomal recessive metabolic disorder caused by a deficiency in the branched-chain alpha-ketoacid dehydrogenase complex, which breaks down branched-chain amino acids [1]. Without prompt treatment, acute metabolic decompensation, often triggered by physical stress, can lead to cerebral edema and death due to the toxic accumulation of leucine. During such a metabolic crisis, timely intervention under the guidance of a metabolic specialist at a tertiary care center is critical. In addition, MSUD is more prevalent in members of plain communities, such as the Amish and Mennonite, due to a founder effect [2]. This case illustrates the critical need for non-metabolic healthcare professionals to be more aware of metabolic emergencies and their management in such populations.

## 2. Case Presentation

A 23-year-old Amish male with poorly controlled MSUD and severe combined immunodeficiency, status post bone marrow transplant, sustained a crush injury to his pelvis and lower extremities at approximately 9 am when a forklift overturned on him during farm work. Emergency medical services (EMS), unaware of the metabolic risks associated with MSUD and without guidelines for metabolic management, transported him by flight to a local community hospital without metabolic specialists. Upon arrival, his open pelvic and femur fractures, and a left groin laceration communicating with the rectum, were promptly treated surgically using Lactated Ringer’s infusion and colostomy creation without metabolic genetics consultation. Despite our emergency transfer request at 12 pm to provide metabolic management, he did not arrive at our tertiary medical center until 7:30 p.m. (~9 h after surgery). We immediately treated him with intravenous dextrose 10% fluids, intralipid infusion, total parenteral nutrition, and MSUD-specific formula. Continuous renal replacement therapy was begun using vas cath placement. Plasma amino acids obtained at 7:30 p.m. revealed elevated levels of isoleucine, leucine, and valine at 217, 876, and 581 μmol/L, respectively (normal ranges: 30–336, 72–201, 119–336, respectively). Creatine kinase level was elevated at 9310 U/L (normal range: 30–200 U/L). About two hours after arrival, he developed sudden sinus tachycardia and hypoxemia, which progressed to bradycardia and ultimately pulseless cardiac arrest of unclear etiology. Despite extensive resuscitative efforts, he passed away 13 h after his injury (Figure 1). Because the family did not request an autopsy, the exact cause of his sudden deterioration remains unclear; potential contributions include cerebral edema, aspiration, embolism, or electrolytic and metabolic derangements.

## 3. Discussion

Acute metabolic decompensation in inherited metabolic diseases can be fatal if treatment is delayed. In conditions such as MSUD, an acute physical stressor–such as traumatic injury–can initiate a cascade of metabolic disturbance that can rapidly perturb the patient’s metabolic homeostasis. Muscle breakdown, commonly seen in crush injuries or major orthopedic trauma, can result in the release of proteins and amino acids, including branched-chain amino acids (BCAAs), into the circulation. In patients with MSUD, impaired degradation of BCAAs can lead to the accumulation of neurotoxic levels of leucine and coma. Additionally, the physiologic stress response to trauma induces a catabolic state, further promoting endogenous protein breakdown. Prolonged fasting, such as perioperative nil per os (NPO) status without dextrose-containing fluids, exacerbates the catabolic drive by depriving the patient of exogenous energy sources and forcing reliance on endogenous protein stores. Required surgical intervention can also exacerbate catabolism due to the combined effects of anesthesia, fasting, tissue injury, and systemic stress. These combined factors can precipitate a catabolic crisis, leading to rapid neurologic deterioration, cerebral edema, and fatal outcome in MSUD.

The primary goal of management is to halt catabolism and promote anabolism. This is achieved by withholding natural protein to limit increases in BCAAs, especially leucine levels, providing caloric intake to prevent further catabolism, including dextrose-containing intravenous fluids (e.g., D10 normal saline), intralipid, metabolic total parenteral nutrition, and specialized metabolic formulas with appropriate levels of BCAAs (Table 1) [1,3]. Lactated Ringer’s infusion is generally avoided due to its potential to cause lactic acidosis and its lack of sufficient dextrose [4]. Another critical aspect of treatment is the prompt removal of toxin metabolites. In MSUD, rapid reduction in plasma leucine levels is essential, typically achieved through continuous renal replacement therapy (CRRT), which requires timely nephrology consultation and placement of a vas cath for CRRT access [5].

Opportunity to manage metabolic emergencies in adults may be relatively limited compared to pediatric cases, while the number of known inherited metabolic diseases has been expanding [6,7,8,9]. Moreover, the management of these metabolic emergencies is often complex, and surgical providers may have not always have ready access to standardized protocols. Given the rarity of inherited metabolic disorders, non-metabolic providers, such as surgical teams and EMS, often have limited clinical experiences. This lack of experience and awareness can contribute to delays in transfer to appropriate metabolic centers. Therefore, increasing awareness of these emergencies, particularly among EMS and adult care teams, is essential. Early recognition and timely intervention guided by a metabolic specialist can be life-saving. Effective management requires a coordinated multidisciplinary team, which includes trauma surgery, medical genetics, nephrology, pharmacy, dietetics, and intensive care members, all playing critical roles. Preparation is key. It is vital to ensure the necessary medical resources are prepared and available quickly, such as D10 normal saline, intralipid, metabolic formulas, and CRRT access. These are all time-sensitive interventions that should be instituted upon patient arrival if clinically indicated.

Individuals in plain communities are at increased risk for metabolic emergencies and delayed treatment due to a combination of genetic, environmental, and sociocultural factors. Genetic factors: It is essential to recognize the elevated prevalence of inherited metabolic diseases in Amish and Mennonite populations, due to the founder effect and high rates of consanguinity in these groups [10]. As a result, these communities have a significantly higher incidence of autosomal recessive conditions, including inborn errors of metabolism, compared to the general population [11]. For example, in the Old Order Mennonite population, the incidence of MSUD is approximately 1 in 358 infants, which is much higher than the global incidence [2]. The estimated carrier frequency for MSUD in the Mennonite community is estimated at 7.96%, indicating that nearly 1 in 12 individuals could transmit the disorder if their reproductive partner is also a carrier [2]. Environmental factors: Occupational and lifestyle-related hazards—such as farm-related duties and the use of equipment, including buggies, farming tools, and machinery—increase the likelihood of traumatic injuries and associated infection [12,13,14]. These injuries can cause wounds, muscle breakdown, rhabdomyolysis, and internal hemorrhage, all of which can trigger catabolism and precipitate a metabolic crisis. In such situations, patients require metabolic management along with urgent surgical intervention. Sociocultural and healthcare access factors: Reduced access to up-to-date metabolic treatment guidelines, protocols, and resources can contribute to delays in treatment [11]. Additionally, cultural practices in some plain communities may discourage the use of modern healthcare interventions, such as immunization and metabolic formulas, or emergency services [11]. Communication barriers, transportation challenges, and limited or delayed contact with tertiary care centers can further limit time-sensitive access to essential and life-saving emergent specialized care [15,16]. To address these barriers, local hospitals and EMS providers should collaborate with metabolic geneticists to become familiar with acute management protocols, and regular education, outreach, and network partnerships are essential to raise awareness and improve preparedness in rural settings.

## 4. Conclusions

In conclusion, this case highlights the importance of early recognition of the need for, availability, and provision of appropriate management of metabolic emergencies in plain communities. Given the increased risk of traumatic injury and systemic barriers to care, non-metabolic healthcare professionals, including EMS providers and trauma teams, must be more aware of the immediate need for and access to specialized treatment for such metabolic emergencies.

The present report is inherently limited by its single-case design. Although it highlights the interplay between trauma, catabolism, fasting, and surgery in precipitating metabolic crisis, these observations may not capture the full spectrum of clinical responses in patients with MSUD or other inborn errors of metabolism. Furthermore, a single-case design limits the generalizability of the findings and precludes conclusions about causality or optimal management. Future research should focus on larger case series or multicenter studies to better define risk factors, outcomes, and therapeutic approaches. In particular, developing standardized protocols for managing metabolic emergencies in adults represents an important area for further investigation.

## Figures and Tables

**Figure 1 reports-08-00201-f001:**
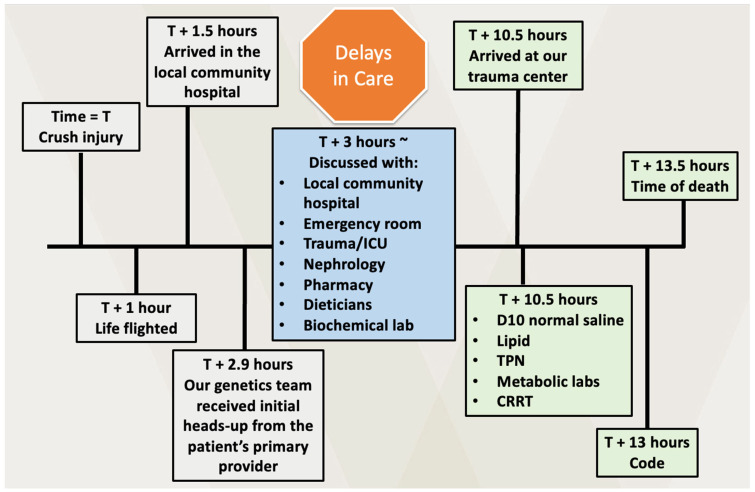
Clinical course and delays in transfer and metabolic treatment.

**Table 1 reports-08-00201-t001:** Acute management strategies for maple syrup urine disorder (MSUD).

Intervention	Rationale
Metabolic genetics consultation	To ensure expert guidance for specific management
Discontinuation of natural protein intake (first 24–48 h)	To halt catabolism and prevent accumulation of branched-chain amino acids (BCAAs), especially leucine
Intravenous dextrose-containing fluids (e.g., D10 normal saline)	To provide calories to promote anabolism, reducing endogenous protein breakdown
Intralipid infusion	To supply additional non-protein calories for anabolic support
Metabolic total parenteral nutrition (TPN)	To maintain nutritional needs while avoiding protein catabolism
Specialized BCAA-free metabolic formulas	To reduce levels of leucine and maintain target levels of BCAAs
Continuous renal replacement therapy (CRRT)	To rapidly reduce levels of leucine

## Data Availability

Data sharing is not applicable as no new data were created or analyzed in this study.

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
