# Peer review of "The Impact of Ethnic and Communication Barriers on Fatal Metabolic Emergent Management of Traumatic Injury: A Case Report"

_reports, 2025, doi:10.3390/reports8040201_

Round 1

Reviewer 1 Report

Comments and Suggestions for Authors

Summary of the Manuscript
The manuscript presents a case report describing the fatal metabolic decompensation of an adult Amish male with maple syrup urine disease (MSUD) who sustained severe traumatic injuries. The report highlights the impact of delayed metabolic intervention, the inappropriate use of standard resuscitation fluids in the absence of metabolic awareness, and the systemic and sociocultural barriers that contributed to the outcome. The main contribution of the article lies in raising awareness of the need for early recognition and prompt multidisciplinary management of metabolic crises in adults with inborn errors of metabolism, particularly within plain communities where such conditions are more prevalent. The integration of clinical, genetic, and sociocultural perspectives provides a valuable discussion that bridges metabolic medicine with emergency and trauma care.

Evaluation of Methodology, Analyses, and Conclusions
As a case report, the methodology is descriptive rather than experimental, but the authors provide sufficient clinical details to reconstruct the sequence of events and treatment decisions. However, the chronology of the patient’s course could be presented more clearly. Timelines, including the time of injury, initial stabilization, surgical intervention, transfer, and initiation of metabolic treatment, would strengthen the reader’s understanding of how delays impacted the outcome. The analyses of laboratory results are appropriate, and the interpretation of elevated branched-chain amino acids is accurate, though more emphasis could be placed on the interaction between trauma-induced catabolism, surgical stress, and the metabolic defect in MSUD. The discussion is generally well supported by literature, but it relies on some older references (e.g., 2003 epidemiological data) that could be updated with more recent genetic studies of Amish and Mennonite populations. The conclusions are valid and emphasize the importance of early recognition and multidisciplinary care; however, they could be strengthened by including more concrete recommendations for EMS protocols, rural hospital preparedness, and culturally sensitive communication strategies.

Constructive Feedback for the Authors

  1. Abstract and Title: Consider restructuring the abstract to provide a concise background, a clear case summary, and a sharper conclusion with specific clinical lessons. The title is somewhat long; a shorter version that emphasizes “fatal metabolic decompensation” and the MSUD-trauma link may attract more readership.

  2. Case Presentation: Provide a more detailed and chronological account of the patient’s management, possibly in table or timeline format, to better illustrate the critical delays in metabolic intervention. Expand on the rationale for treatment choices in the community hospital (e.g., Lactated Ringer’s) and why these are contraindicated in metabolic disorders.

  3. Discussion: Deepen the analysis of the interplay between trauma, catabolism, fasting, and surgery in precipitating metabolic crisis. Broaden the discussion of systemic barriers—transfer delays, lack of hospital protocols, and limited EMS training in metabolic emergencies. Strengthen the sociocultural section by suggesting concrete strategies, such as community education, EMS outreach, or quick-reference guidelines for rural hospitals.

  4. References: Add more recent references (2021–2024) on emergency management of inherited metabolic disorders in adults, as well as updated genetic epidemiology studies in plain communities, to reinforce the novelty of the paper.

  5. Language and Style: Revise the English for clarity and readability. Several sentences are long and complex, and there are minor grammatical issues that could be corrected to improve flow and precision.

  6. Limitations and Ethics: Include an explicit limitations section noting the inherent constraints of single-case reports. Clarify that written informed consent was obtained from the patient’s family for publication.

Author Response

Comment 1
Abstract and Title: Consider restructuring the abstract to provide a concise background, a clear case summary, and a sharper conclusion with specific clinical lessons. The title is somewhat long; a shorter version that emphasizes “fatal metabolic decompensation” and the MSUD-trauma link may attract more readership.

Response to comment 1
We thank the reviewer for these comments. We revised the abstract and title.

Revised abstract

Inherited metabolic disorders can result in fatal metabolic decompensation if not promptly recognized and treated. These conditions are common in Plain communities due to founder effects and the high prevalence of consanguinity. We present the case of an adult Amish male with maple syrup urine disorder who sustained traumatic injuries and delayed metabolic intervention that contributed to fatal outcome. This case highlights the critical need for increased awareness among emergency and adult care providers, especially in Plain communities, and emphasizes the importance of early multidisciplinary coordination and preparedness with metabolic resources to ensure timely, life-saving management in adult metabolic emergencies.

Comment 2

Case Presentation: Provide a more detailed and chronological account of the patient’s management, possibly in table or timeline format, to better illustrate the critical delays in metabolic intervention. Expand on the rationale for treatment choices in the community hospital (e.g., Lactated Ringer’s) and why these are contraindicated in metabolic disorders.

Response to comment 2
We thank the reviewer for these comments. We have created a figure to better illustrate the timeline and emphasize the delay in initiating metabolic intervention. In addition, please see the sentence in the Discussion “Lactated Ringers’ infusion is generally avoided due to its potential to cause lactic acidosis and its lack of sufficient dextrose.”

Added figure

Figure 1. Clinical course and delays in transfer and metabolic treatment

Comment 3

Discussion: Deepen the analysis of the interplay between trauma, catabolism, fasting, and surgery in precipitating metabolic crisis. Broaden the discussion of systemic barriers—transfer delays, lack of hospital protocols, and limited EMS training in metabolic emergencies. Strengthen the sociocultural section by suggesting concrete strategies, such as community education, EMS outreach, or quick-reference guidelines for rural hospitals.

Response to comment 3
We thank the reviewer for these comments. Please see the revised texts.

Revised texts

Acute metabolic decompensation in inherited metabolic diseases can be fatal if treatment is delayed. In conditions such as MSUD, an acute physical stressor–such as traumatic injury–can initiate a cascade of metabolic disturbance that can rapidly perturb the patient’s metabolic homeostasis. Muscle breakdown, commonly seen in crush injuries or major orthopedic trauma, can result in the releases proteins and amino acids, including branched-chain amino acids (BCAAs) into the circulation.  In patients with MSUD, impaired degradation of BCAAs can lead to the accumulation of neurotoxic levels of leucine and coma. Additionally, the physiologic stress response to trauma induces a catabolic state, further promoting endogenous protein breakdown. Prolonged fasting, such as perioperative nil per os (NPO) status without dextrose-containing fluids, exacerbates the catabolic drive by depriving the patient of exogeneous energy sources and forcing reliance on endogenous protein stores. Required surgical intervention, can also exacerbate catabolism due to the combined effects of anesthesia, fasting, and tissue injury, and systemic stress. These combined factors can precipitate a catabolic crisis, leading to rapid neurologic deterioration, cerebral edema, and fatal outcome in MSUD.

Opportunity to manage metabolic emergencies in adults may be relatively limited compared to pediatric cases, while the number of known inherited metabolic diseases has been expanding [6-8]. Moreover, the management of these metabolic emergencies is often complex, and surgical providers may have not always have ready access to standardized protocols. Given the rarity of inherited metabolic disorders, non-metabolic providers, such as surgical teams and EMS, often have limited clinical experiences. This lack of experience and awareness can contribute to delays in transfer to appropriate metabolic centers. Therefore, increasing awareness of these emergencies, particularly among EMS and adult care teams, is essential. Early recognition and timely intervention guided by a metabolic specialist can be life-saving. Effective management requires a coordinated multidisciplinary team which includes trauma surgery, medical genetics, nephrology, pharmacy, dietetics, and intensive care members all playing critical roles. Preparation is key. It is vital to ensure the necessary medical resources are prepared and available quickly, such as D10 normal saline, intralipid, metabolic formulas, and CRRT access. These are all time-sensitive interventions that should be instituted upon patient arrival if clinically indicated.

Sociocultural and healthcare access factors: Reduced access to up-to-date metabolic treatment guidelines, protocols and resources can contribute to delays in treatment. Additionally, cultural practices in some plain communities may discourage the use of modern healthcare interventions, such as immunization and metabolic formulas, or emergency services. Communication barriers, transportation challenges, and limited or delayed contact with tertiary care centers can further limit time-sensitive access to essential and life-saving emergent specialized care. To Address these barriers, local hospitals and EMS providers should collaborate with metabolic geneticists to become familiar with acute management protocols, and regular education, outreach, and network partnerships are essential to raise awareness and improve preparedness in rural settings.

Comment 4

References: Add more recent references (2021–2024) on emergency management of inherited metabolic disorders in adults, as well as updated genetic epidemiology studies in plain communities, to reinforce the novelty of the paper.

Response to comment 4
We thank the reviewer for these comments. Please updated references including PMID: 34829496, PMID: 35313229, PMID: 33817879, and PMID: 36385695.

Updated references

  1. Solares I, Heredia-Mena C, Castelbón FJ, Jericó D, Córdoba KM, Fontanellas A, Enríquez de Salamanca R, Morales-Conejo M. Diagnosis and Management of Inborn Errors of Metabolism in Adult Patients in the Emergency Department. Diagnostics (Basel). 2021 Nov 19;11(11):2148. doi: 10.3390/diagnostics11112148. PMID: 34829496; PMCID: PMC8621113.

  1. Sulaiman RA, Alali A, Hosaini S, Hussein M, Pasha F, Albogami M, Sheikh AN, AlSayed M, Al-Owain M. Emergency management of critically ill adult patients with inherited metabolic disorders. Am J Emerg Med. 2022 May;55:138-142. doi: 10.1016/j.ajem.2022.02.053. Epub 2022 Mar 9. PMID: 35313229.

  1. Ehrenberg S, Walsh Vockley C, Nelson E, Baker J, Arcieri M, Lindenberger J, Ghaloul-Gonzalez L. Under-referral of Plain community members for genetic services despite being qualified for genetic evaluation. J Genet Couns. 2021 Aug;30(4):1084-1090. doi: 10.1002/jgc4.1395. Epub 2021 Apr 4. PMID: 33817879; PMCID: PMC8565999.

  1. Williams KB, Lasarev MR, Baker M, Seroogy CM. Cross-sectional survey on genetic testing utilization and perceptions in Wisconsin Amish and Mennonite communities. J Community Genet. 2023 Feb;14(1):41-49. doi: 10.1007/s12687-022-00621-z. Epub 2022 Nov 17. PMID: 36385695; PMCID: PMC9947211.

Comment 5

Language and Style: Revise the English for clarity and readability. Several sentences are long and complex, and there are minor grammatical issues that could be corrected to improve flow and precision.

Response to comment 5
All sentences in the revised manuscript were reviewed by all authors (native English speakers), to ensure grammatical accuracy and improve readability.

Comment 6

Limitations and Ethics: Include an explicit limitations section noting the inherent constraints of single-case reports. Clarify that written informed consent was obtained from the patient’s family for publication.

Response to comment 6
Please see the Institutional Review Board Statement: “This study was conducted in compliance with the principles of the Declaration of Helsinki. Informed consent for publication of the research details was obtained from the patient's family”. In addition, we have added a limitations section.

Added text

The present report is inherently limited by its single-case design. Although it highlights the interplay between trauma, catabolism, fasting, and surgery in precipitating metabolic crisis, these observations may not capture the full spectrum of responses in patients with MSUD or other inborn errors of metabolism. Additional studies are required to establish standardized management strategies in similar clinical scenarios.

Reviewer 2 Report

Comments and Suggestions for Authors This case addresses a clinically important intersection between inherited metabolic disorders and traumatic injury, especially within socio-culturally distinct populations such as Plain communities. The topic is relevant, timely, and potentially of value to multidisciplinary emergency care providers.   However, before the manuscript can be accepted, I recommend the following revisions to improve its clarity, clinical utility, and conceptual depth: 1. Strengthen the analytical depth of the case narrative. Please expand on the rationale behind key clinical decisions—such as the initial use of Lactated Ringer’s, timing of nephrology consultation, and the 9-hour delay in transfer. Were alternative metabolic strategies attempted at the initial hospital? These clarifications would significantly enhance the educational value of the report. 2. Clarify the cause of death. The patient’s sudden deterioration is described as “unclear,” but readers would benefit from a more robust discussion of possible mechanisms (e.g., cerebral edema, hyperkalemia, embolism, or other metabolic derangements). Was neuroimaging performed? Were electrolytes or gas analyses available? This section currently leaves a critical clinical gap. 3. Expand the discussion of communication and cultural barriers. Given the manuscript’s title, this dimension deserves more than a general paragraph. If applicable, include specifics on family involvement, EMS-provider communication, or cultural hesitations (e.g., use of metabolic formulas, emergency services) that affected care delivery. 4. Consider adding a visual summary or algorithm. A table or figure outlining a proposed protocol for emergency metabolic crisis management in adult MSUD patients (especially for EMS or non-specialist providers) would enhance the manuscript’s practical impact.   I appreciate the authors’ efforts and believe the manuscript will be suitable for publication after addressing the points above.   Decision: Accept after minor revision

Author Response

Comment 1
1. Strengthen the analytical depth of the case narrative. Please expand on the rationale behind key clinical decisions—such as the initial use of Lactated Ringer’s, timing of nephrology consultation, and the 9-hour delay in transfer. Were alternative metabolic strategies attempted at the initial hospital? These clarifications would significantly enhance the educational value of the report. 

Response to comment 1
We thank the reviewer for these comments. At the initial hospital, metabolic management was not initiated because surgery was performed without consultation from metabolic genetics. We have revised the text for clarification.

Added texts

Upon arrival, his open pelvic and femur fractures, and a left groin laceration communicating with the rectum, were promptly treated surgically using Lactated Ringers’ infusion and colostomy creation without metabolic genetics consultation.

Comment 2

  1. Clarify the cause of death.The patient’s sudden deterioration is described as “unclear,” but readers would benefit from a more robust discussion of possible mechanisms (e.g., cerebral edema, hyperkalemia, embolism, or other metabolic derangements). Was neuroimaging performed? Were electrolytes or gas analyses available? This section currently leaves a critical clinical gap.

Response to comment 2
We thank the reviewer for these comments. Neuroimaging, electrolyte testing, gas analysis, and autopsy were not performed; therefore, the exact cause of his sudden deterioration remains unclear. Please see the added texts.

Added texts

Because the family did not request an autopsy, the exact cause of his sudden deterioration remains unclear; potential contributions include cerebral edema, aspiration, embolism, or electrolytic and metabolic derangements.

Comment 3

  1. Expand the discussion of communication and cultural barriers.Given the manuscript’s title, this dimension deserves more than a general paragraph. If applicable, include specifics on family involvement, EMS-provider communication, or cultural hesitations (e.g., use of metabolic formulas, emergency services) that affected care delivery.

Response to comment 3
We thank the reviewer for these comments. Please see the revised texts.  

Added texts

Sociocultural and healthcare access factors: Reduced access to up-to-date metabolic treatment guidelines, protocols and resources can contribute to delays in treatment.  Additionally, cultural practices in some plain communities may discourage the use of modern healthcare interventions, such as immunization and metabolic formulas, or emergency services. Communication barriers, transportation challenges, and limited or delayed contact with tertiary care centers can further limit time-sensitive access to essential and life-saving emergent specialized care. To Address these barriers, local hospitals and EMS providers should collaborate with metabolic geneticists to become familiar with acute management protocols, and regular education, outreach, and network partnerships are essential to raise awareness and improve preparedness in rural settings.

Comment 4

  1. Consider adding a visual summary or algorithm.A table or figure outlining a proposed protocol for emergency metabolic crisis management in adult MSUD patients (especially for EMS or non-specialist providers) would enhance the manuscript’s practical impact.   I appreciate the authors’ efforts and believe the manuscript will be suitable for publication after addressing the points above.

Response to comment 4
We thank the reviewer for these comments. Please see the revised table.  

Round 2

Reviewer 1 Report

Comments and Suggestions for Authors

Congratulations

Author Response

Thank you so much for your positive comment. 

Reviewer 2 Report

Comments and Suggestions for Authors

Thank you for your thoughtful and comprehensive revision. The updated manuscript demonstrates clear improvements in clarity, structure, and clinical relevance.

You have successfully addressed the key areas highlighted during peer review:

  • The case narrative now includes essential clinical rationale and more transparent sequencing of events.

  • The discussion has been expanded to provide a deeper analysis of the pathophysiological mechanisms behind trauma-induced metabolic decompensation.

  • The sociocultural and systemic barriers have been more robustly developed, with constructive suggestions for improving preparedness in rural and Plain community settings.

  • The inclusion of a timeline figure and an updated management table has enhanced the practical utility of the report.

  • You have incorporated recent literature (2021–2024) that reinforces the novelty and relevance of the topic.

The language and formatting have been revised adequately. The limitations section is now explicit and appropriately framed.

This is a valuable case report that will serve as an important reference for emergency care teams, metabolic specialists, and providers working in culturally diverse or resource-limited environments.

No further revisions are required. Congratulations on this important contribution.

Author Response

(The authors gave the same response as above.)
